# Effect of Saline Soil Cracks on Satellite Spectral Inversion Electrical Conductivity

**Xiancong Dong [1,2,3], Xiaojie Li [1,3,*], Xingming Zheng [1,3], Tao Jiang [1,3] and Xiaofeng Li [1,3]**

1   Northeast Institute of Geography and Agroecology, Chinese Academy of Sciences, No. 4888 Shengbei Street, Gaoxinbei District, Changchun 130102, China; dongxiancong@iga.ac.cn (X.D.); zhengxingming@iga.ac.cn (X.Z.); jiangtao@iga.ac.cn (T.J.); lixiaofeng@iga.ac.cn (X.L.)
2   University of Chinese Academy of Sciences, Beijing 100000, China
3   Jingyuetan Remote Sensing Experimental Station, Chinese Academy of Sciences, No. 4888 Shengbei Street, Gaoxinbei District, Changchun 130102, China
*   Correspondence: lixiaojie@iga.ac.cn; Tel.: +86-431-85542224

**Abstract:** The dehydration cracking of saline soil is a kind of common natural phenomenon, and the cracks of saline soil will affect the satellite spectrum, and then affect the accuracy of satellite spectral inversion of electrical conductivity (EC). This study introduces the concept of crack rate (CR) to describe the crack information of saline soil, and quantifies the influence of saline soil crack on the EC of satellite spectral inversion. In 2014 and 2020, the satellite-ground synchronous observation experiments of soda-type inland saline soil and coastal chlorinated-type saline soil were carried out, and the CR of surface cracked saline soil was extracted by an image processing algorithm. For the saline soil spectrum data, the correlation analysis method is used to establish the best band combination that characterizes the relationship between the different saline soil spectrum data and salinity, and the EC inversion model is established using the BP neural network method. The results show that: after the CR is introduced, the determination coefficient ($R^2$) for the EC of soda-type saline soil satellite spectral inversion increased from 0.59 to 0.67, with an increase of 14.42%, and the mean square error (MSE) reduced from 0.20 to 0.16, with a decrease of 19.49%. The $R^2$ for the EC of coastal chlorinated-type saline soil satellite spectral inversion increased from 0.64 to 0.75, an increase of 17.73%, and the MSE decreased from 0.16 to 0.12, a decrease of 25.15%. The study proved the influence of the cracks in the saline soil on the satellite spectrum and provided a new way to improve the accuracy of the satellite spectrum inversion of the EC of the cracked saline soil.

**Keywords:** saline soil; satellite spectra; crack rate; electrical conductivity inversion model; neural network

## 1. Introduction

Soil salinization is a serious natural environmental disaster and it is also the main cause of soil desertification. Especially in arid and semi-arid areas of irrigated farmland grassland and coastal wetlands, the phenomenon of soil salinization is more obvious [1–6]. According to the estimation of the Food and Agriculture Organization of the United Nations (FAO), the global saline soil area is 397 million hectares and the alkaline soil area also amounts to 434 million hectares [7]. In order to ensure the sustainable use of land, more and more attention has been paid to the study of soil salinization [8–14].

Soil salinization refers to the phenomenon or process in which soluble salts accumulate on the soil surface. The soluble salts of saline soil mainly include sulfates, chlorides, carbonates and bicarbonates of sodium, potassium, calcium, and magnesium. Drought, low-lying terrain, poor drainage,

high groundwater level and high salinity of groundwater are the important conditions for the formation of salinization, and the parent material, topography, soil texture also have an important impact on the formation of salinization. The movement of soil salt is mainly governed by the law of soil water movement and the law of salt solubility. Soil salinization leads to the abnormal growth of many type of vegetation, which poses a great threat to agriculture and ecology.

As an important embodiment of the characteristics of saline soil, electrical conductivity (EC) has a direct relationship with the salt content of saline soil, which is a hot research topic among scholars. Remote sensing technology has the advantages of real, objective, timely and accurate, large-scale monitoring, and is widely used in the monitoring of saline soil. Many studies have successfully used remote sensing data to draw soil salinity maps and establish the relationship between satellite spectrum and surface salinization. For example, El Harti et al. [15] proposed a new soil salinity index (OLI-SI), and then they used multi-temporal Landsat Theme Imaging Sensor (TM) and Operational Land Imager (OLI) images from 2000 to 2013 to estimate soil EC to monitor soil salinity in central Morocco, and the $R^2$ of the inversion EC is 0.77. Bai et al. [16] quantitatively estimated soil pH and EC of typical saline-alkali soils in Northeast China based on Landsat 8 land imager (OLI) band data. The soil properties were studied by physical and chemical analysis, statistical analysis, spectral analysis and image analysis, and the variables of soil alkalinity and salinity were determined. Zhang, XG et al. [17] used principal component analysis (PCA) transform, Tasseled Cap (TC) transform and optimal band combination (OBC), three methods to extract information based on the early images of Landsat multispectral scanner (MSS), evaluated the temporal and spatial changes of soil salinization, and analyzed the driving factors of soil salinization. Wang et al. [18] studied the estimation of soil salt content in saline and alkaline soils, applied the continuous projection algorithm to the estimation model, and used the partial least squares regression (PLSR) method to conduct regression modeling. This study proved the applicability of SPA method in estimating soil salinity by using field spectral data and provided a reference for subsequent studies on hyperspectral estimation of soil salinity. Zhang [19] and others, on the basis of UAV multispectral images and Sentinel-2A multispectral images, used EC to carry out satellite UAV ground comprehensive inversion of coastal soil salinity. The spatial and temporal universality of the UAV-based inversion model is verified. The results show that the green, red, red edge and near infrared bands are significantly correlated with soil salinity, and the model's $R^2$ is 0.743 based on the combination of spectral parameters with sensitive bands. Hu et al. [20] used electromagnetic induction (EMI) equipment and a hyperspectral camera installed on a UAV platform to quantitatively characterize and estimate field soil salinity. The results show that the RMSE of a UAV data prediction model is lower than GF-2 data, and the consistency coefficient and performance deviation ratio are higher, which proves the effectiveness of an infrared hyperspectral imager in field soil salinity monitoring and mapping. Sun [21] obtained 1989–2019 Landsat TM/OLI time series data based on a Google Earth Engine platform, and obtained the results of conductivity evolution of saline and alkaline soil in the Songnen plain in the past 30 years based on gaussian process regression model inversion, providing data support for rational planning and utilization of land.

The parameter inversion of saline soil spectral data is a mapping process from multidimensional space to low-dimensional space. This mapping relationship is complex and nonlinear [1]. The BP neural network is an important branch of intelligent computing technology [22–24]. It has the ability to deal with multidimensional and nonlinear data quickly and effectively. The application of a BP neural network to the inversion of saline soil EC can greatly improve the inversion accuracy.

For saline soil, the soil is thick and easy to compact, and cracking occurs during the process of water loss [25,26], as shown in Figure 1. After the saline soil cracked, the soil crust and roughness will change greatly, which will reduce the spectral reflectance of soil and cause the phenomenon of mixed pixels in satellite images. Mougenot et al. [27] have studied different surface morphology and concluded that in general, the reflection of a salinized surface in visible light and the near-infrared band is stronger than that of a general surface, and cracks in crust structure will weaken spectral reflection. For the same salt content of a salt shell, the surface damage will make the roughness increase, and the

overall reflection will decrease, but the spectral curve will not change. Goldshleger et al. [28] found that the crust structure would affect the spectral information of soil texture and soil surface texture, leading to erroneous remote sensing detection results. Therefore, if the influence of salt soil cracking is ignored, there will be a large error in the inversion process of EC. This study introduces the concept of soil CR to describe the crack information of saline soil. The two most common types of salinized soils were selected: inland soda salinized soils and coastal chlorinated salinized soils. In order to verify the accuracy of the model established by the study and make the surface sampling time correspond to the transit time of the satellite, we designed a satellite-ground synchronous observation experiment. The surface CR is extracted by an image processing algorithm. For the saline soil spectrum data, the correlation analysis method is used to establish the best band combination that characterizes the relationship between the saline soil spectrum data and the salinity of different salinities, and the inversion model of EC is established to quantitatively analyze the influence of CR on satellite spectral inversion. The purpose is to provide a new method for improving the accuracy of satellite inversion of saline soil EC.

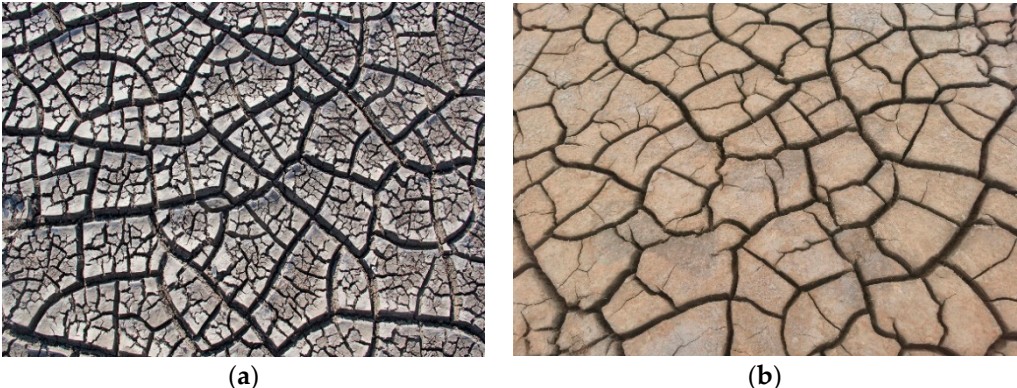

(**a**) (**b**)

**Figure 1.** Cracked saline soil; (**a**) cracking image of saline soil in Songnen Plain; (**b**) cracking image of coastal saline soil cracking.

## 2. Materials and Methods

### 2.1. Study Area and Soil Sampling

This study selected two different saline soil research areas. One is located in Songnen Plain, western Jilin Province, and the soil type is soda saline soil. The other is located in Yingkou City, southern Liaoning Province, and the soil type is coastal chlorinated saline soil.

Songnen Plain is one of the three concentrated distribution areas of soda saline soil in the world, and it is also one of the areas with the most serious secondary salinization and the greatest impact on agriculture in China [29]. The area of salinized land is about 3.42 million $hm^2$, accounting for 20% of the total land area. The study area is located in the hinterland of Songnen Plain of the northwest of Jilin Province, and the geographical location is between 123°47′~124°8′ E and 45°25′~45°33′ N. It belongs to a semi humid continental monsoon climate with four distinct seasons, more wind and less rain, strong evaporation and severe spring drought. The average sunshine hours of the whole year are 3012.8 h, and the annual precipitation is 380–450 mm, and the evaporation is 1100–1500 mm, and water evaporation is about three times the precipitation. It is dry in spring and autumn, rainy and easy to flood in summer, and the precipitation from June to August accounts for 73.1% of the annual precipitation. The topography of this area is low and flat. The surface runoff discharge is not smooth. The pond water surface rises in the rainy season, leading to waterlogging, and the water surface drops or even dries up in the dry season. The special natural background determines the fragility of the ecological environment in this area, that is, the salinization of a flood is serious.

Yingkou City is located on the east coast of the Bohai Sea and the estuary of the Daliao River. The saline soil type is mainly coastal chlorinated saline soil. The study area is located 10 km near the

coastline, and the geographical position is between 122°21′~122°23′ E and 40°32′~40°35′ N. It belongs to a semi-humid continental monsoon climate, with four distinct seasons, rain and heat in the same season. The area has sufficient heat conditions, with annual sunshine hours of 2600~2880 h, annual precipitation of 650~800 mm, accumulated temperature of ≥10 °C of 3300~3670 °C, and dryness coefficient (ratio of possible evaporation during the crop growth period to precipitation during the same period) approximating to 1, which is a special concentrated distribution area of saline-alkali soil in eastern China.

Through field investigations, this study selected experimental area for at least half a month without rainfall time, because of the large evaporation in the experimental area, half a month without rainfall can be thought of as the surface soil moisture is relatively homogeneous, the change of soil moisture tends to be stable. Research collected 158 and 105 cracked soil samples in the two study areas on 23–29 April 2014 and 1–3 July 2020, respectively, as shown in Figure 2. When sampling in the field, the naturally cracked saline soil is selected. Considering the satellite pixel scale, it is required that the CR should be relatively uniform in the 60 × 60 m region to be used as a sampling point, and the locations of each sampling point should be at least 100 m apart. At the same time, at each sampling point, we used a digital camera (Canon 70D, lens model: EF-S 18-135/3.5-5.6 IS) to take vertical photos of the surface at a height of 1 m, and then collected soil samples at a depth of 0–15 cm using a ring knife.

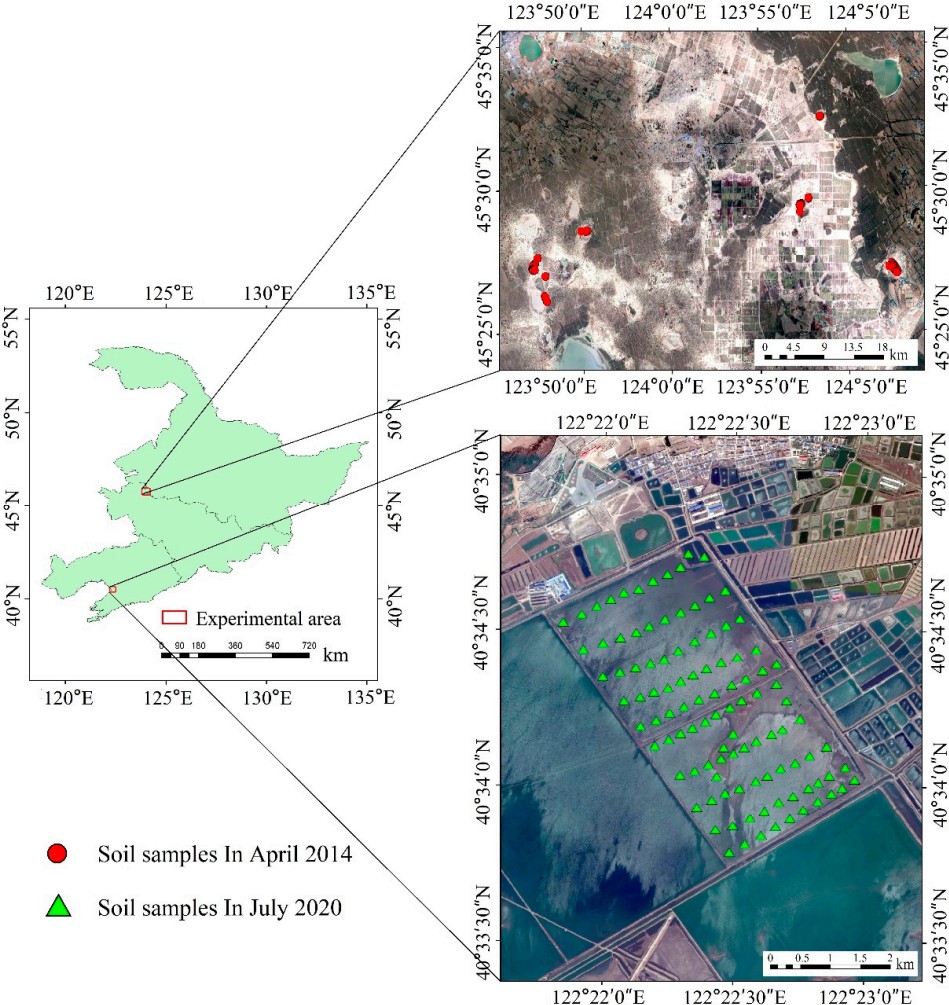

**Figure 2.** The study areas are located in Songnen Plain in the west of Jilin Province of China and Yingkou City on the east coast of the Bohai Sea. The red frame is the experimental sampling area. The solid red circle represents the soil sampling site in the Songnen Plain in April 2014; the green solid triangle represents the soil sampling site in Yingkou City in July 2020.

### 2.2. Remote Sensing Data Source Acquisition and Preprocessing

In order to establish the relationship between ground sampling data and remote sensing image, the transit time of the remote sensing satellite should be consistent with the sampling time of surface soil sample, so as to avoid the influence of weather and surface changes on the research results. In addition, the heterogeneity of surface cracking distribution and salinity distribution also require higher spatial resolution of images. According to the above requirements, the CCD data product of the HJ-1 satellite with spatial resolution of 30 m (http://www.cresda.com/n16/index.html) and the MSI data product of the Sentinel-2 satellite with spatial resolution of 10 m (https://scihub.copernicus.eu/dhus/) were selected respectively. The imaging time is 23rd April 2014 and 30th June 2020. The main parameters of the image are shown in Tables 1 and 2. Sentinel-2 provides 13 bands of multispectral data. Considering the consistency of Sentinel-2 and HJ-1 data and the need of establishing a soil salinity model, the red, green, blue and nir bands of the Sentinel-2 image were selected, and the image was resampled to 30 m resolution by cubic convolution interpolation. All four bands were selected for the HJ-1 image. The data product provided by the Sentinel-2 satellite is Level-1C, which is the apparent reflectivity of the upper atmosphere after radiation correction and geometric correction. Each Level-1C product is composed of an orthographic image with a scene of 100 km$^2$, and the map coordinates of the image are corrected by a digital elevation model (DEM), including data such as land, water and cloud masks. The Sen2cor software, provided by the European Space Agency (ESA), was used to calibrate the atmosphere of Sentinel-2 satellite images to generate the Level-2A data product. The data product provided by the HJ-1 satellite is a level-1 product for system geometric correction. ENVI software is used for radiometric calibration and FLAASH atmospheric correction according to satellite parameters. The reflectance of the sampled band is obtained.

**Table 1.** Spectral information of Sentinel-2 satellite bands.

| Sentinel-2 Bands | Central Wavelength (μm) | Resolution (m) |
| --- | --- | --- |
| Band 1—Coastal aerosol | 0.433 | 60 |
| Band 2—Blue | 0.490 | 10 |
| Band 3—Green | 0.560 | 10 |
| Band 4—Red | 0.665 | 10 |
| Band 5—Vegetation Red Edge | 0.705 | 20 |
| Band 6—Vegetation Red Edge | 0.740 | 20 |
| Band 7—Vegetation Red Edge | 0.783 | 20 |
| Band 8—NIR | 0.842 | 10 |
| Band 8A—Vegetation Red Edge | 0.865 | 20 |
| Band9—Water | 0.945 | 60 |
| Band 10—SWIR—Cirrus | 1.375 | 60 |
| Band 11—SWIR | 1.610 | 20 |
| Band 12—SWIR | 2.190 | 20 |

**Table 2.** Spectral information of HJ-1 satellite bands.

| HJ-1 Bands | Wavelength Range (μm) | Resolution (m) |
| --- | --- | --- |
| Band 1—Blue | 0.430–0.520 | 30 |
| Band 2—Green | 0.520–0.600 | 30 |
| Band 3—Red | 0.630–0.690 | 30 |
| Band 4—NIR | 0.760–0.900 | 30 |

The construction of spectral parameters includes two methods: one is generated by the combination operation of sensitive bands (that is, addition and multiplication operation) [30]; the other is to evaluate the degree of soil salinization by using various existing spectral indexes. Combined with the existing research results, due to the high correlation between soil salinity and EC, this study selected the following spectral indices and EC for correlation analysis, including the soil salinity index SI1, SI2, SI3,

normalized salt index (NDSI), and ratio salt index (SI-T) [31–33]. The calculation formula of each index is shown in Table 3. The calculation formula of the correlation coefficient is:

$$R = \frac{\sum_{i=1}^{n}(x_i - \overline{x})(y_i - \overline{y})}{\sqrt{\sum_i^n(x_i - \overline{x})^2}\sqrt{\sum_i^n(y_i - \overline{y})^2}} \tag{1}$$

where $x_i$ is the spectral index, $\overline{x}$ is the average spectral index, $y_i$ is the EC, $\overline{y}$ is the average EC.

**Table 3.** Spectral index.

| Spectral Indices | Formula |
|---|---|
| Soil salinity index 1 | $SI1 = \sqrt{G \times R}$ |
| Soil salinity index 2 | $SI2 = \sqrt{G^2 + R^2 + NIR^2}$ |
| Soil salinity index 3 | $SI3 = \sqrt{G^2 + R^2}$ |
| Normalized salinity index | $NDSI = (R - NIR)/(R + NIR)$ |
| Ratio salinity index | $SI - T = R/NIR$ |

### 2.3. Measurement of EC

There are three ways to measure EC in soils: measuring pore water EC, bulk EC, or saturation extraction EC. The three ways are related in some way, but there are tools to convert one into the other. In this study, our object is cracked soil, and there will be a big error if we use bulk EC, so we chose saturation extraction EC. The collected soil samples were air-dried to eliminate the effects of moisture, then they were grounded until they could pass through a 2 mm sieve. For each soil sample, a soil suspension with a water-to-soil ratio of 1:5 [34] was prepared using distilled water. After standing at 25.0 °C for 24 h, the EC of the soil solution was measured by using a lightning magnetic EC meter (model DDS-307A), which is recorded as $EC_{1:5}$.

### 2.4. CR Extraction Algorithm Based on Image Processing

At each sampling point, we used a digital camera to take pictures of the soil surface vertically at a height of 1 m above the ground. The shooting time and sampling time were synchronized, the weather was clear, and the contrast of the photos met the precision of CR extraction in the laboratory. A digital image processing method was used to extract the CR of the cracked soil. Figure 3 takes one sample as an example to illustrate the major steps. First, the color image obtained with the digital camera in Section 2.1 was converted to a gray image (Figure 3b), and a threshold was chosen based on the gray histogram for binarization (Figure 3c), then inversion was performed (Figure 3d). As a result, the sample's image was segmented into aggregate parts (indicated by black pixels) and cracks. Erosion was then performed on the binary image to remove noise pixels (Figure 3e). For this image, CR can be calculated by this formula:

$$CR = \frac{A_C}{A_S} \tag{2}$$

where $A_C$ is the total white pixels, and $A_s$ is the total pixels of the image.

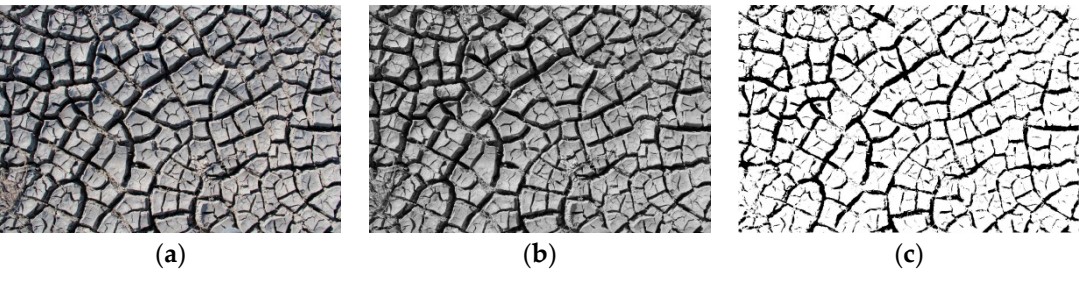

(a)       (b)       (c)

**Figure 3.** *Cont.*

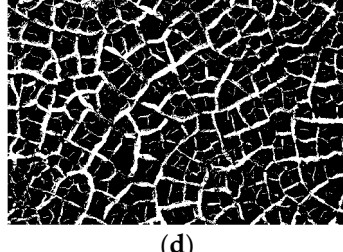
**(d)**

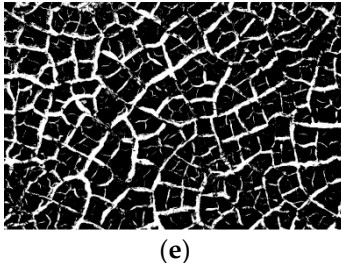
**(e)**

**Figure 3.** Image processing steps. (**a**) original image; (**b**) gray image; (**c**) binarization; (**d**) inversion; (**e**) remove noise.

### 2.5. Construction of CR Model

Affected by the degree of salinity, the vegetation cannot grow on the naturally cracked saline soil, which is covered by pure bare soil. Therefore, the information contained in a single satellite pixel can be approximately divided into two parts: soil and cracks. That is, it is assumed that the information value of any pixel is the linear weighted sum of the reflectivity $r_0$ of the crack part and the reflectivity $r_1$ of the non-crack part:

$$r = r_0 + r_1 \tag{3}$$

In a pixel containing only soil and cracks, it is assumed that the crack coverage ratio on the surface corresponding to the pixel is *CR*, and the corresponding soil coverage ratio is 1-*CR*. It is assumed that the information contribution value of pure soil pixels is $S_{soil}$, and the information contribution value of non-soil pixels is $S_{CR}$. Then, the information value of the whole mixed pixel is:

$$r = CR \times S_{CR} + (1 - CR) \times S_{Soil} \tag{4}$$

Theoretically speaking, in the actual saline soil cracking area, the light wave has been reflected many times in the crack and finally received by the sensor with very little energy and presents a black shadow on the optical image. Therefore, in this study, the reflectivity of the crack contribution is approximately regarded as zero, that is, $S_{CR} = 0$. The reflectivity of the whole cell should be:

$$r = (1 - CR) \times S_{Soil} \tag{5}$$

The spectral reflectance of satellite images with *CR* is calculated by using the above formula.

### 2.6. Inversion Model for Electrical Conductivity of Cracked Saline Soil

Neural network is a popular machine learning algorithm that has proven its effectiveness in using satellite images to predict and evaluate various soil physical parameters such as soil moisture [35], soil salinity [3], and soil organic carbon [36]. The BP neural network uses a back propagation algorithm training. The basic idea is that if the expected output cannot be obtained by forward propagation using the weight and threshold, then back propagation, repeatedly modify (iterate) the weight and threshold of each node, and gradually reduce the small cost function until it reaches the preset requirements. Generally, when the cost function is less than a fairly small positive number or the iteration is no longer reduced but oscillated repeatedly, the training of the network is completed and the mapping relationship between the input and output of the model is determined. In short, the weights are adjusted to minimize the total error of the network. The training process of the BP neural network is divided into two stages: forward propagation and backward propagation [37]:

Forward propagation stage: (1). Obtain a sample $(X_p, Y_p)$ from the sample set and input $X_p$ into the network; (2). Calculate the corresponding actual output $O_p$.

Backward propagation stage (error propagation stage): (1). Calculate the difference between the actual output $O_p$ and the ideal output $Y_p$; (2). Adjust the weight matrix in a way to minimize the error;

(3). The error of the network about the p-th sample measure: $E_p = \frac{1}{2}\sum_{j=1}^{m}\left(y_{pj} - O_{pj}\right)^2$; (4). The error measure of the network about the whole sample set: $E = \sum_p E_p$.

In this study, a BP neural network model was established to retrieve the EC of saline soil. The salinity index and band reflectance, which have a significant effect on soil EC were selected as the input layer of the neural network, and the output layer was set as a node, corresponding to the saline soil EC. *Tansig* function was selected as the transfer function of the hidden layer, and *purelin* linear neuron was selected as the output layer, so that the output of the whole network could take any value. In this study, *traingdm* (gradient descent method with additional momentum), *traimlm* (L-M optimization algorithm) and *Trainbr* (Bayesian regularization algorithm) were trained respectively. It is found that the convergence speed of the *traingdm* function is too slow, while that of the *trainlm* function is fast, but it is easy to produce transition matching. The convergence speed of *Trainbr* function lies between them, and the precision of the cyclic test is also high. Therefore, this study used a Bayesian regularization algorithm to build the BP neural network model. Since the dimensions of the input data are different from those of the output data, the sample data should be normalized by the *premnmx* function first, and then denormalized by the *postmnmx* function after training. According to a certain proportion, the model randomly divides the sample point set into the training set, verification set and test set. The training set presents errors to the network during the training process and adjusts the network based on its errors. The validation set is used to measure network generalization, and stops training when generalization stops improving. The test set has no direct effect on network training and is an independent measure of network performance during and after network training. The model structure is shown in Figure 4.

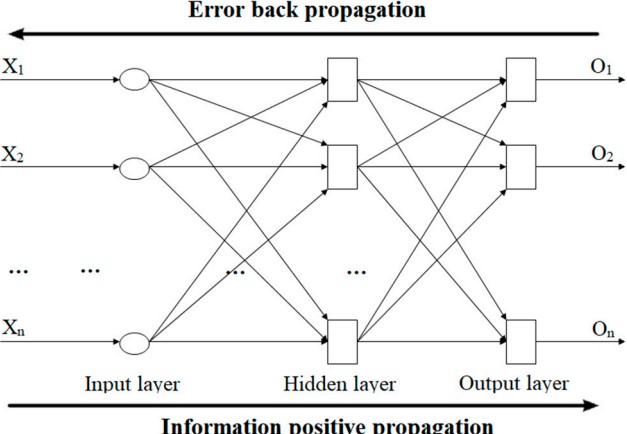

**Figure 4.** Schematic diagram of BP neural network model.

The Matlab R2014a software was used to compile the BP neural network code. In order to obtain the optimal training model, the model was set so that when the training performance is reduced to a minimum or the performance gradient is lower than minimum level, it will automatically stop and perform a self-test. According to the prediction results and accuracy of the model, we adjusted the model parameters and the dimensionality of the neural network, and the number of hidden layer neurons was set as 50, the minimum target error was 0.0001, and the maximum training times were 10,000. The MSE and R$^2$ on the validation and test sets were then estimated to obtain the parameter values that have the highest degree of fit to the sample set to construct the model.

### 2.7. Model Evaluation Indexes

In this study, *MSE* and $R^2$ were selected as the evaluation indexes for model accuracy. MSE refers to the expected value of the square of the difference between the estimated and the true values of the

parameter. It can evaluate the degree of data change, and the smaller the value of MSE is, the better the description accuracy of experimental data of the prediction model. The calculation formula is:

$$MSE = \sum_{i=1}^{n} (f(x_i) - y_i)^2 \tag{6}$$

where $f(x_i)$ is the measured EC, and $y_i$ is the estimated EC of the model.

$R^2$ is the coefficient of determination, also known as the goodness of fit. It is used as a statistical indicator to reflect whether the change of the dependent variable is reliable. The higher the $R^2$ is, the better the effect of the model is. The calculation formula is:

$$R^2 = \frac{SSR}{SST} = 1 - \frac{SSE}{SST} \tag{7}$$

where *SST* is the total sum of squares, and *SSR* is the regression sum of squares, and *SSE* is the sum of squares of residuals (residual: the difference between the actual value and the observed value).

## 3. Results

### 3.1. Descriptive Statistics for the Soil EC

Statistical results of $EC_{1:5}$ of saline soil in two study areas in this study are shown in Table 4.

**Table 4.** Descriptive statistics of saline soils $EC_{1:5}$ (unit: dS/m).

|  | Number | Maximum | Minimum | Mean | Median | SD | CV (%) |
|---|---|---|---|---|---|---|---|
| Songnen Plain | 158 | 4.20 | 0.13 | 1.81 | 0.70 | 0.92 | 50.83 |
| Yingkou City | 105 | 86.00 | 24.45 | 50.76 | 49.55 | 14.01 | 27.60 |

SD is standard deviation; CV = SD × 100/mean; EC is electrical conductivity.

It can be seen from Table 4 that the distribution range of $EC_{1:5}$ in Songnen Plain is from 0.13 to 4.20 dS/m, indicating that the soil in the study area is affected by soil salinization. The average value of $EC_{1:5}$ is 1.81 dS/m, and the median value is 0.70, indicating that most sampling areas in this study area are moderately salinized. The SD and coefficient of variation (CV) were 0.92% and 50.83%, respectively, which indicated that the degree of salinization in the sampling area was moderately discrete, but the variability was large.

The distribution range of $EC_{1:5}$ in Yingkou City is from 24.45 to 86.00 dS/m, which indicates that the soil salt content in this area is extremely high and is greatly affected by seawater. The average value of $EC_{1:5}$ is 50.76 dS/m, and the median value is 49.55, indicating that the salinization degree in this study area is extremely severe. The SD and CV are 14.01 and 27.60%, respectively, which indicates that the degree of salinization in this region is discrete, but the variability is not strong and the degree of salinization is uniform.

The main reason for the great difference of soil EC between the two research areas is the different geographical location. The saline soil in Songnen Plain is mainly affected by soil parent material and evaporation far higher than precipitation. Most of the surface salt accumulation belongs to surface evaporation reverse salt. The Yingkou research area is close to the ocean, and the groundwater level is only one to five meters. Seawater is poured and transported a large amount of salt, resulting in extremely high salt content in this area.

### 3.2. Descriptive Statistics for the Soil CR

Statistical results of CR of saline soil in the two study areas in this study are shown in Table 5.

**Table 5.** Descriptive statistics of saline soils crack rate (CR).

|  | Number | Maximum | Minimum | Mean | Median | SD | CV (%) |
|---|---|---|---|---|---|---|---|
| Songnen Plain | 158 | 0.14 | 0.00 | 0.05 | 0.03 | 0.04 | 89.33 |
| Yingkou City | 105 | 0.16 | 0.01 | 0.08 | 0.05 | 0.03 | 37.15 |

SD is standard deviation; CV = SD × 100/mean.

The statistical results of CR of cracked soil calculated according to Section 2.4 are shown in Table 5. In the Songnen Plain, of 158 samples, the lowest CR is 0 and the highest CR is 0.14. The reason for the zero CR of soil cracking may be due to the low salt content of the surface soil and the thin surface cracks of the soil, which can be ignored. The average CR is 0.05, the median is 0.03, and the CV is 89.33%, which indicates that most of the sampling areas are affected by salinization to varying degrees and show cracking characteristics.

The statistical data of CR in the coastal saline soil of Yingkou City show that the maximum and minimum value are 0.16 and 0.01, respectively, which is similar to the cracking of the Songnen Plain. The average CR and median of the sampling points are 0.08 and 0.05, respectively, indicating that the CR in this area is generally higher than that of the Songnen Plain. It can be seen from Section 3.1 that the salt content of coastal saline soil is higher than that of the Songnen Plain, which is in line with the trend that the CR of saline soil increases with the increase of salt content [35]. The CV is 37.15%, indicating that the crack degree changes smoothly in this area, which is related to the uniformity of the sampled areas.

### 3.3. Selection of Sensitive Band and Salinity Index

In this study, the correlation analysis between the measured EC data and the satellite band spectral reflectance and the spectral index SI1, SI2, SI3, NSDI, and SI-T was carried out ($p < 0.01$). The correlation coefficient (R) values are shown in the following Table 6, and the index with high |R| and significant correlation is selected as the parameters of the inversion model.

**Table 6.** Pearson correlation analysis between EC of saline soil and spectral band and salinity index.

|  | Blue | Green | Red | NIR | SI1 | SI2 | SI3 | NDSI | SI-T |
|---|---|---|---|---|---|---|---|---|---|
| Songnen Plain | −0.128 | −0.135 | −0.075 | −0.013 | −0.109 | −0.058 | −0.101 | −0.308 ** | −0.306 ** |
| Yingkou City | 0.241 | 0.279 ** | 0.321 ** | 0.336 ** | 0.305 ** | 0.339 ** | 0.308 ** | −0.102 | −0.095 |

**: At the 0.01 level, the correlation is significant.

Generally speaking, if the R is greater than or equal to 0.3, we hold that there is a good linear correlation between variables. Therefore, NDSI and SI-T are selected as the salinity indices of the saline soil EC inversion model in the Songnen Plain, and SI1, SI2 and SI3 are selected as the salinity indices of the coastal saline soil EC inversion model in Yingkou City.

It can be seen from Table 6 that the band reflectivity and salinity index of HJ-1 have a negative correlation with EC, and have a good correlation with the NSDI and the SI-T. Sentinel-2 band reflectance and salinity index are mostly positively correlated with EC, and have a good correlation with salinity index SI1, SI2, and SI3. The reason for the different positive and negative correlations between the band reflectivity and EC of the HJ-1 satellite and the Sentinel-2 satellite is mainly caused by the difference in the salt type of the saline soil. The sensitivity of band reflectance to different soil salinity types is different, and then the correlation coefficient is affected. In addition, the HJ-1 satellite and Sentinel-2 satellite have obvious opposite correlations to the five salt indices. By comparing the formulas of five salinity indices in Table 3, it is found that the main difference between SI1, SI2, SI3 and NDSI, SI-T is

that the former three indices include Green band, while the latter two do not. As for the effect of the Green band, this will be the direction to be considered in the next research.

### 3.4. Satellite Spectral Inversion Model of Saline Soil EC

The EC inversion model of saline soil based on the BP neural network realizes highly nonlinear mapping from multidimensional information of the soil spectrum to EC. The macroscopic difference of saline soil spectra is not obvious, but the BP neural network can identify and distinguish the subtle differences between them. The BP neural model was established by using satellite spectral data, soil salinity index and soil $EC_{1:5}$.

### 3.4.1. Inversion of EC of Saline Soil in the Songnen Plain

In order to make the model prediction more accurate and objective, this study randomly divided the data of 158 sampling points into three parts according to the ratio of 70:15:15, and selected 112 sampling points as the training set, 23 sampling points as the verification set and 23 sampling points as the test set to construct the neural network model. The inversion accuracy of the model is shown in Figure 5. The $R^2$ of the model inversion results and the measured EC is 0.59 and MSE is 0.20. In the range of 0~2 dS/m, a few of the estimated values are too high, which is due to the relatively low salt content in this part of the sample area on which a small amount of saline vegetation grows, which increases the satellite spectral reflectance, resulting in a small amount of estimated values being too high. In the range of measured EC > 2 dS/m, the estimated value is low, which is due to the fact that the sample area is located near the pond, and the soil water content is high, which reduces the reflectivity of the satellite band.

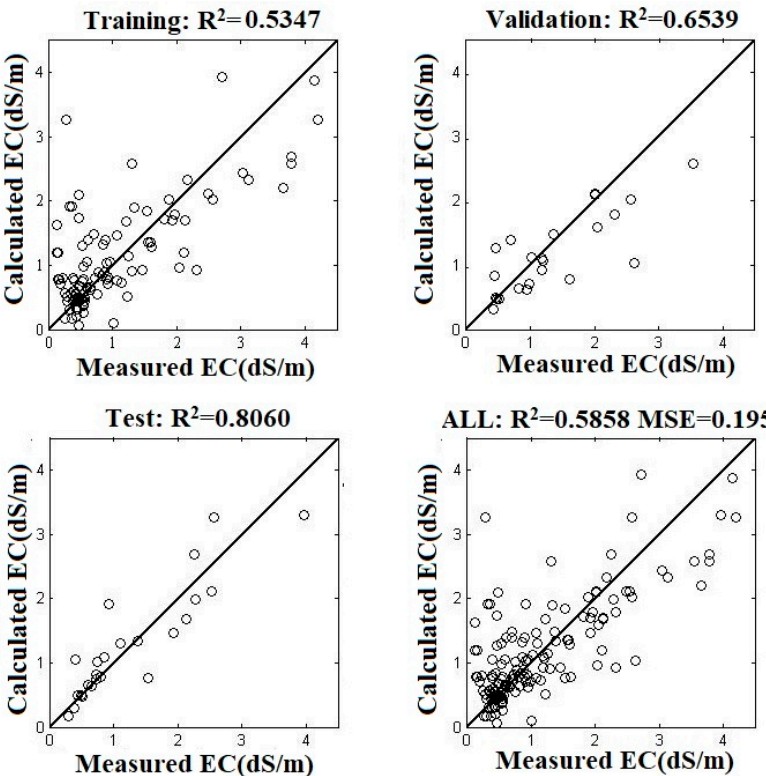

**Figure 5.** The accuracy of the BP neural network model to invert the EC value and to measure EC in soda saline soil.

Then, we used the CR model built in Section 2.5 to calculate the satellite spectral reflectance and salinity index after introducing the CR, and built the BP neural network model. Similarly, the sample points are divided into training set, validation set and test set according to the ratio of 70:15:15.

The running result is shown in Figure 6. The $R^2$ of the model inversion result and the measured EC is 0.66, and the MSE is 0.16. Compared with the result before CR was introduced, $R^2$ increased by 14.42% and MSE decreased by 19.49%. It can be seen from the figure that the estimated value of the model after introducing CR is closer to the 1:1 line. Regardless of the 0~2 dS/m interval of the measured EC or the interval >2 dS/m, the deviation of the model fitting value is obviously reduced.

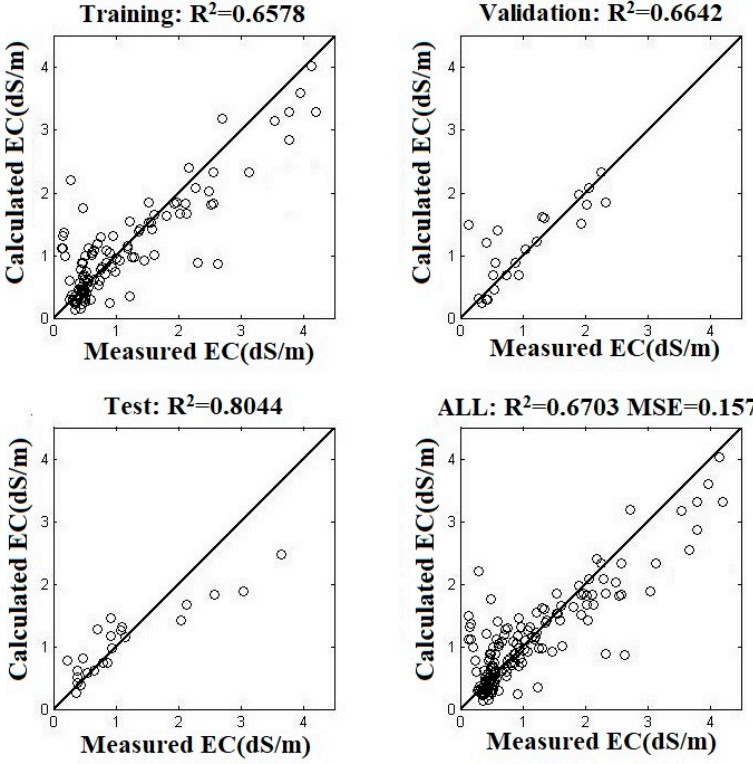

**Figure 6.** After the CR is introduced, the accuracy of the BP neural network model to invert the EC value and to measure EC in soda saline soil.

### 3.4.2. Inversion of EC of Saline Soil in Yingkou City

According to the ratio of 70:15:15, the research randomly divides the data of 105 sampling points in the Yingkou coastal area into three parts: training set, verification set and test set, and constructs the BP network model. The accuracy of the model's inversion result is shown in Figure 7. The $R^2$ of the model's inversion result and the measured EC is 0.64, and the MSE is 0.16. Except for one or two allowed outliers, the model fitting values fit well on both sides of the 1:1 line, which is related to the relatively uniform sampling area.

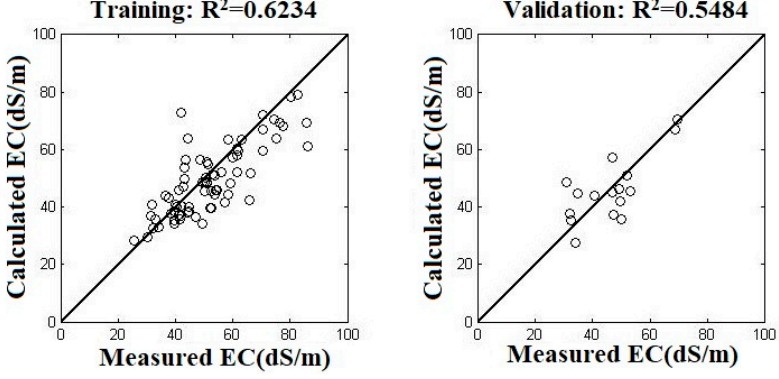

**Figure 7.** *Cont.*

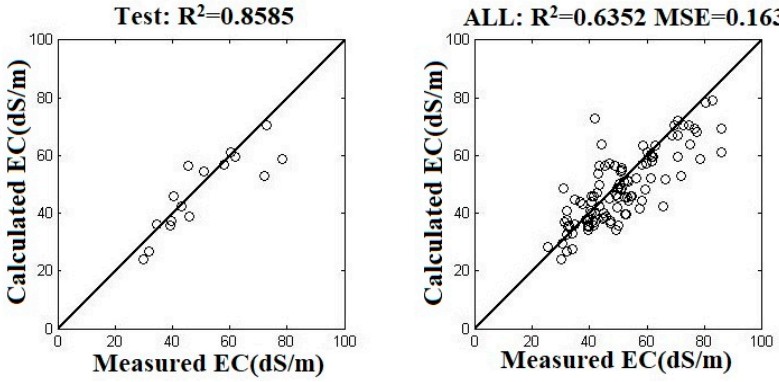

**Figure 7.** The accuracy of the BP neural network model to invert the EC value and to measure EC in coastal saline soil.

Then, we used the CR model built in Section 2.5 to calculate the satellite spectral reflectance and salinity index after introducing the CR, and built the BP neural network model. The running result is shown in Figure 8. The $R^2$ of the model inversion result and the measured EC is 0.75, and the MSE is 0.12. Compared with before CR was introduced, $R^2$ increased by 17.73% and MSE decreased by 25.15%. The fitting value of the model after introducing CR is closer to the 1:1 line, and the result shows high fitting accuracy. It is proved that the introduction of CR can greatly improve the inversion of satellite spectral EC of cracked saline soil. Table 7 shows the accuracy of EC inversion before and after the introduction of CR in two research areas.

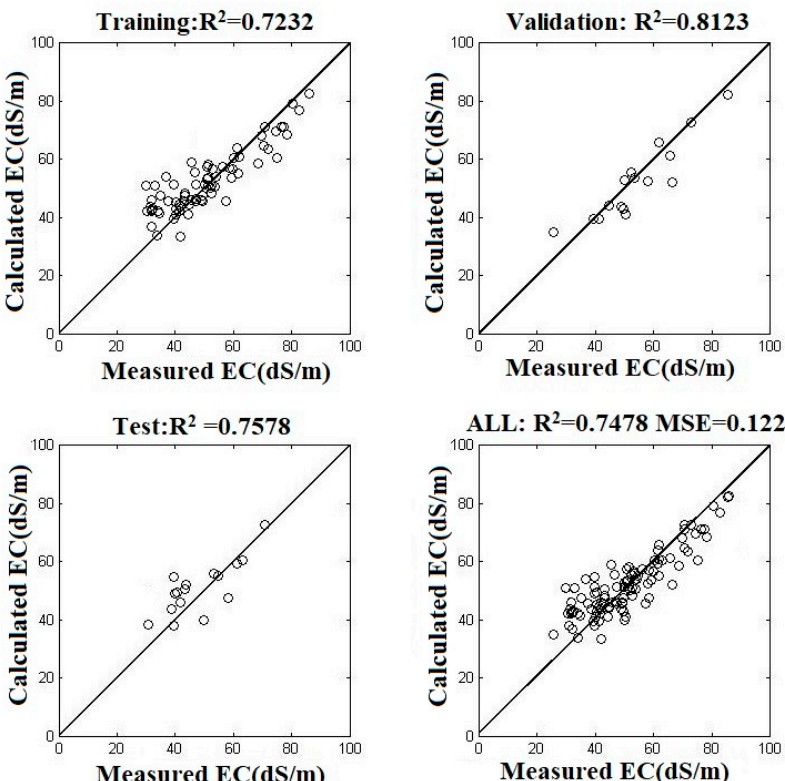

**Figure 8.** After the CR is introduced, the accuracy of the BP neural network model to invert the EC value and to measure EC in coastal saline soil.

**Table 7.** The accuracy of the model before and after the CR is introduced.

| Evaluation | Before | | After | | Improve Rate/(%) | |
|---|---|---|---|---|---|---|
| | MSE | $R^2$ | MSE | $R^2$ | MSE | $R^2$ |
| Songnen Plain | 0.20 | 0.59 | 0.16 | 0.67 | 14.42 | 19.49 |
| Yingkou City | 0.16 | 0.64 | 0.12 | 0.75 | 17.73 | 25.15 |

## 4. Discussion

Soil cracking is a complex mechanical process, which is affected by a series of factors such as soil texture, soil type and soil moisture. During the formation of saline soil, a large amount of water in the surface soil evaporates [38–40], and the upward capillary movement of soil water is stronger than infiltration and gravity movement. Soluble salt compounds accumulate on the soil surface with the evaporation and condensation of water. In the process of water loss from soil moisture, bound water film will be formed between soil particles with high salt content, which reduces cementation and increases the distance between soil particles, resulting in the decrease of soil cohesion and tensile strength [41], shrinkage cracking and cracks. After the saline soil cracks, the soil crust and roughness will change greatly, which will reduce the spectral reflectance of soil and make the satellite image produce mixed pixel phenomenon. Therefore, as an important factor affecting the satellite spectrum, CR needs to be considered in the construction of the inversion model.

In this study, the soil samples collected in the field were treated by air drying, grinding and sieving, and the soil EC was measured, and the mapping model was established with satellite spectral data. On the one hand, soil EC can reflect the content of salt ions in soil; on the other hand, the response characteristics of different salts to spectra are different, which provides a basis for retrieving soil EC according to spectral characteristics. For the spectral information obtained by satellite, the relationship between soil spectrum and EC of saline soil is reflected without considering the influence of satellite data errors such as atmosphere, solar altitude, terrain fluctuation and satellite sensor performance. However, these errors are real. Although the satellite image is corrected by radiation and atmosphere, there are still some errors between the corrected band value and the real surface value. This mapping relationship is complex and nonlinear. The BP neural network is an important branch of intelligent computing technology, which has the ability to process multidimensional and nonlinear data quickly and effectively. In this paper, the BP neural network is used to retrieve the EC of saline soil, which greatly improves the retrieval accuracy of saline soil EC.

In this study, a Sentinel-2 image in the Yingkou coastal research area was resampled by convolution interpolation three times, and the resolution of the image was resampled from 10 to 30 m. On the one hand, due to the fact that the land surface is uniform and the sampling points are evenly distributed in the study area, resampling the image can reduce the heterogeneity of the land surface, thus making the prediction accuracy of EC higher. On the other hand, it can be consistent with HJ-1 satellite resolution, and it is more convenient to observe the difference of EC inversion of different saline soil types under similar band range and band resolution.

It can be seen from 3.4 that the fitting accuracy of the BP neural network model has been significantly improved through the CR model proposed and constructed in this study, and the influence of soil cracks on satellite spectrum has been eliminated to some extent. However, the CR model constructed by the research regards the crack reflectivity as approximately zero value, which is lower than the actual crack reflectivity, and it will affect the final inversion accuracy. In the follow-up study, the CR model can be further improved.

According to the descriptive statistics analysis of variance performed for soil EC and CR in the two research areas, it can be seen that there is a great difference in EC when the CR in the two study areas is similar. This is related to different salt types in the study area. Different soil salt types will affect the composition and structure of soil, and then have a certain impact on the occurrence of cracks. The specific influencing process and reasons will be carried out in the following research.

Different soil moisture will also affect the retrieval of satellite spectra and EC. This study focuses on the influence of soil CR, without taking the soil moisture into account, which can be taken as the next improvement point.

The cracking of saline soil is very common in nature. The CR model proposed in this study can greatly improve the accuracy of satellite spectral EC inversion of cracked saline soil, and quantitatively analyze the influence of CR on satellite spectrum inversion. It provides a new method for improving the accuracy of satellite inversion of the EC of saline soil. At the same time, the proposed method can be applied to the inversion of various soil parameters of cracked saline soil, which brings a new promotion to the inversion of soil elements such as moisture and organic matter of saline soil. In addition, the cracking structure and texture characteristics of cracks may be closely related to soil texture and salt types, and the exploration of these problems may bring a new direction to the retrieval of soil parameters by satellite remote sensing. However, the CR in this study was obtained by field photography, and the obtaining method was relatively troublesome. How to obtain a wide range of crack information more quickly and simply to provide support for research is still a big challenge.

## 5. Conclusions

In order to reduce the influence of saline soil cracking on satellite inversion and improve the accuracy of satellite inversion of saline soil, in this study, the concept of soil CR was first proposed to describe the crack information of saline soil, and two different types of saline soil were selected for validation. The results show that: after introducing the CR model proposed in this study, the $R^2$ for the EC of soda-type saline soil satellite spectral inversion increases by 14.42%, and the MSE decreases by 19.49%; the $R^2$ for the EC of coastal chlorinated-type saline soil satellite spectral inversion increases by 17.73%, and the MSE decreases by 25.15%. Whether it is the soda-type saline soil in the Songnen Plain or the coastal chlorinated saline soil in Yingkou City, the fitting accuracy of the BP neural network model is significantly improved. In the pixels with only soil and cracks, the CR is an important variable that affects the satellite spectrum. Effectively eliminating the influence of CR is of great significance not only for the inversion of soil EC, but also for the retrieval of other physical parameters of saline soil. In general, the CR model proposed in this study will provide support for effective monitoring of cracked saline soil, and the performance of the model will also be improved by combining other important environmental factors.

**Author Contributions:** Conceptualization, X.D. and X.L. (Xiaojie Li); methodology, X.D. and X.Z.; writing—the original draft preparation, X.D.; writing—review and editing, X.D., X.L. (Xiaojie Li) and X.Z.; experiment, X.D., X.L. (Xiaojie Li) and T.J.; supervision, X.L. (Xiaojie Li), X.Z., X.L. (Xiaofeng Li), T.J.; project administration, X.L. (Xiaojie Li); funding acquisition, X.L. (Xiaojie Li) All authors have read and agreed to the published version of the manuscript.

**Funding:** This research was funded by the National Natural Science Foundation of China (No. 41671350 and No. 41771400); National Key Research and Development Project of China (No. 2019YFC0409101).

**Conflicts of Interest:** The authors declare no conflict of interest.

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
