# Peer review of "Effect of Saline Soil Cracks on Satellite Spectral Inversion Electrical Conductivity"

_remotesensing, doi:10.3390/rs12203392_

Round 1

Reviewer 1 Report

  1. Statements mentioned in line 75-77 is important for the manuscript and need to support by references

        Line 75-76 “This mapping relationship is complex and nonlinear.

        Line 76-77 “BP neural network is an important branch of intelligent                  computing technology. It has the ability to deal with multidimensional              and nonlinear data quickly and effectively“

  1. It is not clear reasoned why soda and chlorinated saline soils were selected to find different CR effect on soil salinity assessment. Necessary to add more information (Line 91).
  2. It is also not clear necessity “design satellite-ground synchronous observation experiments”. Give more explanation (Line 92).
  3. It was not full explained and understandable why in the study soil samples collected 158 and 105 cracked soil samples (Line 130).
  4. It is also will be good for readers to put connection from which study area number of samples were taken respectively (Line 130-131);
  5. It is also need to be explained why dates on April 23-29, 2014 and July 1-3, 2020 were chosen (Line 131);
  6. It is not enough explained process of removing noise pixels (Line 188);
  7. CR ration calculated on the base of experimental data analysis and applied for certain known areas. Does it mean that remotely assessed saline areas with possibly cracked areas need to be always support by field experiments?
  8. It is necessary to add and  show results of ANOVA.
  9. Discussion part written in style of introduction part and need to be improved.

Author Response

Dear Reviewer:

Thank you for your review and Suggestions on our manuscript 《Effect of Soil Cracks on Satellite penetration inversion Electrical Conductivity(ID: Remotesensing-918618)》.
Those comments are all valuable and very helpful for revising and improving our paper, as well as the important guiding significance to our researches. We have studied comments carefully and have made correction which we hope meet with approval.Please see the attachment.

Best regards

Authors

Reviewer 2 Report

The title of the manuscript (MS) deals with "Effect of Saline Soil Cracks on Satellite Spectral inversion Electrical Conductivity". The topic of this manuscript is of interest and well written and I liked reading it, great job!

Just one comment.

In the "Introduction" section, there is a need that you will use "recent publications" on the topic to make attractive your research. In this section, speaking about the development of modern technology and the advantage of using Remote Sensing (RS) technology, the authors should provide several references to substantiate the claim made in this section (that is, provide references to other groups who do or have done research in this topic) to make the introduction more substantial for example GEE; https://rb.gy/ww6kuu, etc...

Author Response

Dear Reviewer:

Thank you for your review and suggestion of our manuscript.《Effect of Soil Cracks on Satellite penetration inversion Electrical Conductivity(ID: Remotesensing-918618)》.

We are very happy that our work has been approved by you. We have also adopted your suggestions and revised the manuscript, hoping to get your next approval.Please see the attachment.

Best regards

Authors

Reviewer 3 Report

The manuscript “Effect of Saline Soil Cracks on Satellite Spectral inversion Electrical Conductivity” is within the scope of the journal and addresses a topic relevant to a broad audience.

The abstract is not clear or particularly informative. It misses critical information that is relevant to understand the scope and results presented in the manuscript.

There is no mention to the environmental conditions at sampling and in the period leading up to the field campaigns. Soil moisture can have a very important impact in the measured and estimated variables and must be considered.

Although the Crack Ratio is calculated in a way not described entirely in the literature it is also not as new as suggested. The analysis of RGB imagery, converted to gray scale and segmented into a binary product was described before in papers as old as 2016.

The image collection method is also not entirely clear and should be described in greater detail (for instance, does the time of day and different shadows/contrast influence the results?)

Line 155-156, it is mentioned that radiometric and atmospheric corrections were applied. How? This is important. Sentinel-2 is available as a level-2 product. Was that used?

It is also not clear how many samples per satellite pixel were collected. Only one and assumed the pixel is homogeneous? That is hardly a sound approach and must be clarified.

BP neural networks are a potentially good choice for the problem at hand. However, there is not information on how it was implemented (which environment, language, libraries). It is also unclear whether the hyperparameters were fine-tuned. A lot more information is needed.

The discussion should also be expanded as it does not delve deep into the challenges and opportunities created by the method.

Figure 2 should be improved. A scale in the insets is required

Author Response

(The authors gave the same response as above.)

Reviewer 4 Report

The paper in question is a very well written manuscript and it covers the subject of the research thoroughly. Concerning the salinity aspect in this paper the approach of the authors is sufficiently documented. The language used is scientific and the technologies described are very well documented. Both the salinity and the remote sensing approaches are very interesting and adds to the current knowledge. I have no major comments or suggestions to make to the authors. Keep up the good work.

Author Response

Dear Reviewer:

Thank you for your review  on our manuscript 《Effect of Soil Cracks on Satellite penetration inversion Electrical Conductivity(ID: Remotesensing-918618)》.
We are very happy that our work has received your approval.wishing you good health and everything goes your way.

Best regards

Authors